# The Prevalence of *Encephalitozoon cuniculi* in Domestic Rabbits (*Oryctolagus cuniculus*) in the North-Western Region of Romania Using Serological Diagnosis: A Preliminary Study

**DOI:** 10.3390/microorganisms12071440

**Published:** 2024-07-16

**Authors:** Anca-Alexandra Doboși, Anamaria Ioana Paștiu, Lucia-Victoria Bel, Dana Liana Pusta

**Affiliations:** 1Department of Genetics and Hereditary Diseases, Faculty of Veterinary Medicine, University of Agricultural Sciences and Veterinary Medicine Cluj-Napoca, 400372 Cluj-Napoca, Romania; anca.dobosi@student.usamvcluj.ro (A.-A.D.); dana.pusta@usamvcluj.ro (D.L.P.); 2New Companion Animals Veterinary Clinic, Faculty of Veterinary Medicine, University of Agricultural Sciences and Veterinary Medicine Cluj-Napoca, 400372 Cluj-Napoca, Romania; lucia.bel@usamvcluj.ro

**Keywords:** *Encephalitozoon cuniculi*, encephalitozoonosis, domestic rabbit, serology, ELISA

## Abstract

*Encephalitozoon cuniculi* is a microsporidian, domestic rabbits being the main host. The disease can be acute or subclinical, but treatment options are limited and usually with unrewarding results; therefore, diagnosis and prevention of encephalitozoonosis in rabbits are of the utmost importance. This study aims to obtain the first preliminary information of the prevalence of *E. cuniculi* in the north-western region of Romania. A total of 176 rabbits were clinically examined and 2 mL of blood was sampled from each. An enzyme-linked immunosorbent assay (ELISA) kit by Medicago (Medicago, Uppsala, Sweden) on the resulted blood serum was utilized. Statistical analysis of the results was conducted using the EpiInfo 2000 software (CDC, Atlanta, GA, USA). A total prevalence of 39.2% (69/176) was identified, with statistically significant differences in relation to the rabbits’ clinical status, age, season of sampling, breeding system, body condition score and county of origin; the different family farms tested also had a statistically significant difference. This study gives the first preliminary information on this pathogen distribution on Romania’s territory, but further studies need to be performed on larger regions to declare the prevalence in the country.

## 1. Introduction

*Encephalitozoon cuniculi* is a eukaryotic, unicellular, spore-forming, obligate intracellular pathogen of the phylum Microsporidia [1]. The main host of this microorganism is the domestic rabbit (*Oryctolagus cuniculus*), but it has been widely identified in other species such as dogs, cats, birds, mice, monkeys, foxes and even humans [2]. For the latter, infection usually occurs in patients suffering from AIDS or other immunodeficiencies, with a wide range of symptoms [1,2]. In rabbits, two main pathways of transmission have been identified: the horizontal route, through excretions contamination or spore inhalation, and the vertical or transplacental route, from doe to kit [3]. Potential risk factors that can lead to infection of the rabbits are household rearing compared to commercial farms, due to a less controlled sanitary system; the use of multi-animal enclosures that favors the contamination of the rabbits with urine and feces; diets comprising fruit, vegetables or grain; and foods that possess a higher potential of contamination [4]. The main organs affected by this pathogen have been documented to be the brain, kidney and eyes, but it has also been previously located in the liver and the heart [5]. Therefore, clinical symptoms expressed in the rabbit are mainly neurological, such as head tilt, ataxia, nystagmus, paresis, tremors, seizures with or without rolling and hind limb paralysis with urinary incontinence [6,7,8]. Renal symptoms described are polyuria, polydipsia and insufficiency, but are usually hard to be observed [8]. The quite specific lesion of the eye is phacoclastic uveitis [7]. 

Diagnosis methods include mainly histopathologic, serological and molecular genetics techniques, with a variability regarding the specificity and sensibility of these [3]. Histopathology using different types of stains usually reveals lesions of granulomatous meningoencephalitis and chronic interstitial nephritis, as well as the presence of spores in the targeted organs. Diagnosis using molecular genetic techniques is another type of method that is attempted more and more for *E. cuniculi* diagnosis, such as nested PCR (polymerase chain reaction) and real-time PCR, which have proven to be the most successful in DNA detection of the microorganism, but the sensitivity of this method is still being questioned [9]. Ante-mortem diagnosis of encephalitozoonosis using serology remains to be the standard method that can help assess if there has been an exposure of the animal to the pathogen, or more exactly, to exclude the infection in the case of repeated negative results [10]. It is important to always interpret the results of this test in relation to the clinical state of the rabbit, since *E. cuniculi* cannot always be established as the causative agent of the disease, and high antibody titers can persist also in an asymptomatic state, even for years. Generally, the titers of the specific IgM and IgG antibodies are determined as part of the organisms’ humoral immune responses, where IgMs are usually ascending in an acute infection, the elevation of both IgM and IgG simultaneously indicate an active infection (acute, reactivated infection or reinfection) and the detection of specific IgGs alone signals a chronic or latent infection [11]. The serological techniques used are enzyme-linked immunosorbent assay (ELISA), indirect fluorescent antibody test (IFAT), carbon immunoassay (CIA), Western blot analysis and C-reactive protein (CRP) measurement [12]. Quantitative titer detection can be implemented in the ELISA technique, but most methods usually work only on qualitative antibody detection. Some studies showed a good correlation between the qualitative results of ELISA and IFAT and between IFAT and CIA [13], as well as between ELISA and CIA [14].

Treatment of the disease in rabbits is quite problematic because there is no cure that guarantees a full recovery of the patient. The medication usually attempted is fenbendazole orally dosed at 20 mg/kg body weight daily for 28 days, but also together with steroidal anti-inflammatory drugs, broad-spectrum systemic antibiotherapy, sedatives, antiemetics, topical ocular drugs and supportive therapy [3,15], depending on the severity of the case. Prevention of infection is a very important aspect that needs to be taken into consideration, since acute disease can lead to fatality. This can be accomplished by periodical serological testing of rabbit populations, prophylactic fenbendazole administration [8] and thorough disinfection of the environment [3]. These methods, however, still do not ensure that *E.cuniculi* is completely cleared, and constant monitorization of rabbit populations needs to be performed.

In the area of Romania, *E. cuniculi* has never been officially reported, therefore no information on the prevalence of this microorganism is known. This is most likely due to economical issues, regarding the cost of diagnosis in relation to the animal value, as well as the lack of large rabbit farms in the present. The rabbit industry, especially for pet animals, is currently ascending, which is why the introduction of this diagnosis as a standard method is very important to detect and control the disease level.

The aim of this study was to serologically test rabbit populations originating from different breeding facilities in the north-western region of Romania, thus obtaining the first preliminary information about the prevalence of encephalitozoonosis.

## 2. Materials and Methods

### 2.1. Animals and Sampling

Between June 2022 and January 2024, blood samples were collected from 176 rabbits, from both pets and family farms located in the north-western region of Romania. The pet rabbits (*n* = 28) were presented at the Clinic of New Companion Animals of the Faculty of Veterinary Medicine of Cluj-Napoca, Romania, and the rabbits destined for meat consumption (*n* = 148) originated from family farms of the counties Bistrița-Năsăud, Cluj, Satu-Mare and Sălaj, Romania. All the counties of origin and the number of rabbits tested from each are highlighted in Figure 1. Each animal was clinically examined and information regarding breed, age, sex, season of sampling, weight, BCS (body condition score), vaccination status and health status were recorded.

The breeds of the rabbits sampled were Californian, Continental Giant, Dwarf Rex, Flemish Giant, French Lop, Holland Lop, Hycole, Lionhead, Rex, Transylvania Giant, Vienna Blue and mixed breeds. Considering the age, they were categorized using the model of another study [16] as young (≤4 months old) and adult (>4 months old), where 46 were young rabbits and 130, adults. Regarding the sex of the rabbits, 87 were males, 82 females and 7 of unidentified sex due to young age. As for the season of the blood sampling, 35 were recorded during spring (March-May), 57 during summer (June-August), 17 during autumn (September-November) and 67 during the winter (December-February) season. The weight of the animals was recorded between 600 g and 9 kg. BCS was interpreted on a scale from 1 to 5, where 1 represents severe underweight, 3 is a normal weight and 5 is an obese animal [17]. The animals taken into this study were categorized in just 2/5, 3/5 and 4/5 BCS, based on their weight and clinical aspect at the moment of sampling. Generally, the vaccination in rabbits may be carried out against RHD (rabbit hemorrhagic disease) and myxomatosis. Out of the total number of 176 rabbits, 107 were vaccinated against RHD 1 ± 2 and/or myxomatosis, while 69 rabbits were completely unvaccinated. 

Clinical status was categorized as either symptomatic, showing signs specific to encephalitozoonosis (neurologic, renal or ocular symptoms), or asymptomatic, where rabbits were either clinically healthy or showing non-specific symptoms (dental, otic, gastrointestinal, respiratory or cutaneous diseases). Among the pet rabbits, 16 of them were symptomatic, mainly presenting neurological signs such as ataxia, nystagmus, head tilt or even seizures with rolling, while the other 12 showed no specific signs of the disease. The rabbit populations originating from family farms (*n* = 148) were all categorized as asymptomatic. 

Pet rabbits were kept inside the household, either free-range or in an enclosure, with low contact to other rabbits or other animals. Family farm rabbits were mainly kept outside, either in multi-rabbit enclosures or single cages, thus with a higher risk of contamination from other rabbits or species from the surrounding area. These data were not recorded for each individual and taken into analysis due to the variability of this factor, e.g., pet rabbits still having a potential contact with other animals and outside conditions by going into the veterinary facility for medical care. 

The sampling according to the number of the rabbits tested and the correlation with the rabbit data, such as age, sex, season of sampling, breeding system, BCS, vaccination status, clinical status and county of origin are presented in Table 1.

Approximately 2 mL of blood was collected from each rabbit by using the lateral saphenous vein with a 23 G needle and a clot activator blood collection tube. Each sample was centrifuged at 1500 rpm for 10 min and sera was collected and stored at −18 °C before testing.

Sample collection was performed with the written consent of the owners. The study was approved by the Animal Ethics and Welfare Committee of the University of Agricultural Sciences and Veterinary Medicine, Cluj-Napoca, Romania (No. 320/3 June 2022).

### 2.2. Serological Analysis

All sera collected were tested for antibodies anti-*E. cuniculi* using a commercial indirect enzyme-linked immunosorbent assay (ELISA, Medicago^®^, Uppsala, Sweden), following the manufacturer’s instructions. Serum samples were diluted 1:40 with phosphate buffer saline (PBS), while the positive and negative controls were diluted 1:100 with PBS and the conjugate, 1:1000 with PBS, all prior to dispensing into antigen-coated plates. All samples were analyzed in replicates, as the instructions required introducing each sample in an *E. cuniculi*-coated well and in a control antigen-coated well, respectively. Absorbances (A) were read at 450 nm. ELISA results were interpreted using the manufacturer’s protocol (Sample A450 *E. cuniculi*-coated/Sample A450 control antigen-coated). Obtained values ≥ 2.1 were classified as positive for *E. cuniculi* antibodies and values ≤ 2.0 were classified as negative for *E. cuniculi* antibodies. 

### 2.3. Statistical Analysis

Data of the rabbits (age, sex, season of sampling, breeding system, BCS, vaccination status, health status and county of origin) were analyzed using Pearson’s chi-squared test for independence, using the *E. cuniculi* serological status (positive/negative) as the dependent variable. Confidence Intervals (CI) for each of the variables were calculated using EpiTools 0.5-10.1 (Sergeant, ESG, 2018) (http://epitools.ausvet.com.au, accessed on 15 May 2024). Significance was set at *p* < 0.05. Statistical analysis was performed using the EpiInfo 2000 software (CDC, Atlanta, GA, USA) and Prism^®^ 10 (Graph Pad Software, San Diego, CA, USA) software.

## 3. Results

A total prevalence of 39.2% (69/176, 95% CI: 32.30–46.57) was determined, with statistically significant differences depending on age, season of sampling, breeding system, BCS, clinical status and county of origin, as shown in Table 1 and Appendix A. 

The risk factors related to *E. cuniculi* out of the statistical analysis seem to be the age, where adults had a significantly higher prevalence of 21.74% (59/130; 95% CI: 12.26–35.57%); the blood sampling season, where springtime had the significantly highest prevalence of 71.43% (25/35; 95% CI: 54.95–83.67%); the breeding system, where the pet category had a significantly greater seropositivity of 57.14% (16/28; 95% CI: 39.07–73.49%); the BCS, where the 2/5 score presented the biggest prevalence of 60% (3/5; 95% CI: 23.07–88.24%); the clinical status of the rabbit, where the symptomatic rabbits clearly showed a significantly higher seropositivity of 68.75% (11/16; 95% CI: 44.40–85.84%); and the county of origin, where Bistrița-Năsăud came out with the significantly higher seroprevalence of 58.33% (7/12; 95% CI: 31.95%).

Since testing in the family farms was performed as a screening method in asymptomatic rabbits destined for meat consumption, we wanted to analyze these facilities individually and identify the statistical relevance of our results, as presented in Table 2.

The seropositivity among the family farms ranged from 0% (95% CI: 0–56.15%, farm A; 95% CI: 0–16.11%, farm H) to 87.50% (95% CI: 52.91–97.76%, farm D) from the total of 11 facilities analyzed; a statistical significance was recorded (*p* = 0.0008).

## 4. Discussion

In the present, studies on the diagnosis of *E. cuniculi* in rabbits through serological methods have been performed several times, on rabbit categories such as pet, farm or laboratory. The relation between the clinical status and the seropositivity can vary, depending on the rabbit showing specific signs of encephalitozoonosis, non-specific symptoms or a complete asymptomatic state. The serological methods mainly used by other authors are ELISA, IFAT or CIA.

In this study, 69 out of 176 samples (39.2%) came out positive for *E. cuniculi* antibodies using the ELISA method, which proves the presence of the pathogen among the rabbit population of north-western Romania. This serological method was used because of its relatively high sensitivity. The rabbits included in the study not only originated from two types of breeding systems, but were also in different clinical states, which gave us the opportunity to analyze the variability of pathogen identification. The specific types of antibodies present (IgM or IgG) and their titers could not be identified, therefore the moment of infection or the current infection status could not be determined, but the sole detection of the antibodies gives us enough information to state that this pathogen has a wide distribution, and that the risk of contamination with rabbit excretions and vertical transmission has an important role. 

Regarding the rabbits’ age, the young group had a 21.74% (10/46) prevalence and the adult group, a 45.48% (59/130) prevalence, which proved to have a significant difference. Even though the groups were not equally distributed, this statistical significance is supported by other studies [16,18], where rabbits older than 4 months also had a higher prevalence in comparison to the younger rabbits. This can be explained by the fact that maternal antibodies are transmitted from doe to kit and can be present up to 4 weeks of age, time in which they will test seropositive, and then from 4 to 8 weeks of age they will turn seronegative, due to antibody disappearance [18]. In the present study, positive results were also identified in the 4 to 8 weeks-old timeframe, but this can be explained through the possible transplacental route of transmission and the higher prevalence in the adult group, which would lead to an infection from doe to kit. 

The relation of seropositivity to the rabbits’ sex in this study was not significant, which was also discovered in other studies [8,18,19]. Vaccination status was also the only other variable which did not show a statistical relevance to *E. cuniculi* antibody detection, giving us an idea that immunity is specific to RHD and myxomatosis and seems to play no role in the organism’s capacity to have a better immune response against encephalitozoonosis.

A relevant statistical difference obtained in this study was in the relation between the season of sampling and seropositivity, with the spring and autumn seasons having the higher prevalences of 71.43% (25/35) and 52.94% (9/17), respectively, compared to summer with 33.33% (19/57) and winter with 23.88% (16/67), respectively. Another author [20] reported no statistically significant differences regarding the season of sampling for *E. cuniculi* antibody detection in rabbits, but a study [21] on a similar microsporidian, *Enterocytozoon bieneusi* in cattle, revealed similar results to this study, with higher prevalences in the spring and autumn seasons. This can be explained by favorable temperature and rainfall conditions in these seasons, which allow the spores to survive and be transmitted easier, compared to summer and winter, where extreme temperatures impair the microsporidian growth.

The breeding system also proved to be a significant variable in relation to seropositivity, where pet rabbits had a 57.14% (16/28) prevalence and rabbits from family farms, a 35.81% (53/148) prevalence. The higher prevalence in the pet category could be explained by the fact that, in this group, symptomatic cases were present, whereas the family farm group consisted exclusively of asymptomatic rabbits. The two groups, however, are not equally distributed for an accurate comparison of prevalences. It has been previously reported [22] that rabbits from commercial farms had a relevantly lower seropositivity compared to rabbits from family farms, explained through better hygiene conditions present in the industrial system. 

The BCS in this study was evaluated on the 1 to 5 scale, but all rabbits included were fit only between the 2–4 scores. A significant difference was also observed here in relation to seropositivity, where the 2/5 and 4/5 had higher prevalences of 60% (3/5) and 46.88% (15/32), respectively, while the rabbits with a normal BCS of 3/5 had a 36.69% (51/139) prevalence. These results could suggest that an organism which is not balanced in terms of weight could be more susceptible to a poorer immune system against *E. cuniculi* and disease, but additional studies would be required to support this theory.

The symptomatic rabbits of this study came out to be seropositive in a 68.75% (11/16) percentage, while the animals showing no specific clinical signs had a lower prevalence of 36.25% (58/160), the association of this variable with seropositivity coming out as significantly different. Even if the groups were not equally distributed, other studies [23,24] support the same results of a higher seroprevalence in clinically ill rabbits compared to healthy ones, thus showing the high possibility of *E. cuniculi* as the causative agent of disease manifested through neurological, renal or ocular signs. The concern about the seropositivity in the asymptomatic group is related to the fact that 148 out of the 160 rabbits were animals destined for meat consumption and a zoonosis is in discussion. One study [25] performed on murine models by using fermented pork meat infected with *E. cuniculi* demonstrated that infected meat could be a potential transmission source to humans. Further studies still need to be carried out to analyze the true risk of rabbit meat contamination on human consumption.

As an additional analysis of the seropositivity in relation to the pet group and their clinical status, 68.75% (11/16) of the symptomatic pet rabbits had a positive result, while a lower percentage of 41.66% (5/12) of the asymptomatic pet animals had *E. cuniculi* antibodies present. The higher prevalence in the symptomatic group goes to show that rabbits manifesting neurologic signs, as previously described, have a high chance of having *E. cuniculi* as the causative agent of the disease, which should help the clinician in establishing a presumptive diagnosis and a further treatment and prophylactic protocol.

Analyzing the different counties of Romania from which the rabbits originated, we obtained a statistically significant difference, with prevalences ranging from 0% (0/20) to 58.33% (7/12). The highest number of rabbits in the study came from the county of Cluj, where the prevalence obtained was 50.52% (49/97), but this is also where most of the symptomatic rabbits came from. In the other counties tested, animals usually came from one and the same family farm, therefore further studies on multiple farms from the same region would be necessary to assess the real prevalence.

By analyzing the prevalences of each family farm tested, as shown in Table 2, we can see the variety of the *E. cuniculi* distribution, having farm H with a 0% (0/20) seropositivity, to farm D with almost a full positive status of 87.50% (7/8). Even if the rabbit populations sampled from each farm were not equally distributed, a statistically significant difference was still obtained, which lays the foundation for future studies based on larger and equally distributed rabbit populations in this same region of Romania.

To evaluate the general *E. cuniculi* distribution worldwide, other research must be noted as follows: while in Southern Italy seroprevalences of 32% [16] and 67.2% [18] were reported, slightly higher percentages of 59.5% [23] and 70.5% [26] were identified in the Northern and Central Italy. A lower seropositivity of 18.5% [27] in rabbit farms was identified in the region of Sardinia, while testing for *E. cuniculi* antibodies in hares by the same author revealed a 0% prevalence. Other studies reported different seroprevalences of 71.4% [19] in Slovenia; 36.2% [22] in Czech and Slovak Republics; 59% [5] and 52% [28] in the United Kingdom; 43% [29] in Germany; and 29% [30] in Finland. A more in-depth study carried out in Austria [10], that categorized tested pet rabbits into three groups, identified a 70% seropositivity in rabbits with specific symptoms for encephalitozoonosis, a 50% seropositivity in rabbits with non-specific symptoms and a 100% seropositivity in the individuals with phacoclastic uveitis, which also proved the high possibility of *E. cuniculi* being responsible for causing neurologic, renal and ocular signs. Going further on the other continents, detection of *E. cuniculi* antibodies has been reported to be 63.5% and 28.5% for IgG and IgM antibodies, respectively, in Japan [31]; 63% and 67%, using the CIA and ELISA technique, respectively, noted in Taiwan [14]; 22.6% [32] in Korea; 21.9% [33] and 16.22% [20] in China; 49.5% [34] in Turkey; 82% [8] in Brazil; and 62% [24] in the United States of America. 

The high prevalences cited by other authors [5,8,14,18,19,24,26,28] highlight the importance that should be put on reducing the contamination risks and increasing the prophylactic measures. Although the type of food consumed by rabbits has been reported to possess a risk for *E. cuniculi* infection due to potential contamination [4], there are studies that showed that food is not among the risk factors for *E. cuniculi* infection [30]. Thorough disinfection of the environment and periodic serological testing of rabbits remain the main techniques for diminishing the spread of *E. cuniculi* in rabbit populations worldwide.

The mean prevalence value obtained in this study (39.2%) is the first one reported in any region of Romania; therefore, it is helpful to assess the first preliminary insight on the microsporidian spread on this territory. This study comprised both pets and family farm rabbits, which not only presents us with the epidemiological situation in these different types of breeding systems, but also helps local veterinarians in shaping a therapy and prophylactic plan for further control of this microorganism’s spread. Introducing a screening method for the standard diagnosis of *E. cuniculi* is hoped to be accomplished in the future, but the economical aspect also needs continuous development in order for this to happen. In terms of future study directions, further research should be carried out on larger populations from all over the country, to establish the real prevalence and to be able to correctly compare it to other countries.

## 5. Conclusions

The present study identified the seropositivity of rabbit populations along the north-western region of Romania, providing the first information on *E. cuniculi* presence in the territory of this country. The prevalence obtained was 39.2%, but considering that the biggest portion of the rabbits tested were asymptomatic and destined for meat consumption, it raises concerns about the amount of prophylactic measures applied, as well as about the zoonotic potential of the disease. The limitations of this study include a relatively low number of individuals tested and the reduced area of the country involved, as well as a low number of symptomatic rabbits compared to asymptomatic ones. Further studies are expected to be carried out on larger populations and in other counties, in order to have a clearer image of the microsporidian spread among rabbits and how this can be diminished.

## Figures and Tables

**Figure 1 microorganisms-12-01440-f001:**
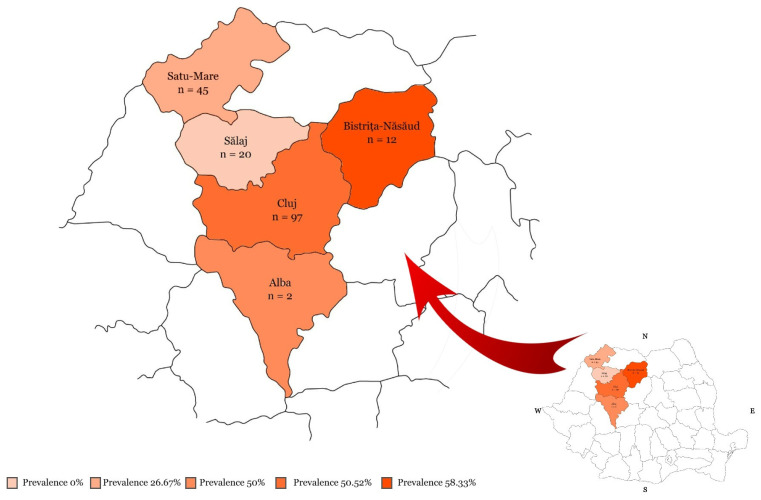
The counties of origin and the number of rabbits tested for *Encephalitozoon cuniculi* in the north-western region of Romania. The shade of color is in accordance with the resulted prevalence.

**Table 1 microorganisms-12-01440-t001:** *E. cuniculi* prevalence based on ELISA testing performed on 176 rabbits from the north-western region of Romania.

Rabbit Data	No. Rabbits Tested	No. Positive Rabbits	Prevalence (%)	CI 95%	*p* *
Age					0.0047
Young (≤4 months)	46	10	21.74%	12.26–35.57	
Adults (>4 months)	130	59	45.38%	37.08–53.95	
Sex					0.7532
Male	87	33	37.93%	28.45–48.43	
Female	82	34	41.46%	31.42–52.27	
Unidentified	7	2	28.57%	8.22–64.11	
Season of sampling					0.00001
spring	35	25	71.43%	54.95–83.67	
summer	57	19	33.33%	22.49–46.28	
autumn	17	9	52.94%	30.96–73.83	
winter	67	16	23.88%	15.27–35.33	
Breeding system					0.0339
Pet	28	16	57.14%	39.07–73.49	
Family farm	148	53	35.81%	28.54–43.80	
BCS					0.0003
2/5	5	3	60%	23.07–88.24	
3/5	139	51	36.69%	29.14–44.96	
4/5	32	15	46.88%	30.87–63.55	
Vaccination status					0.0648
vaccinated	107	36	33.64%	25.40–43.03	
unvaccinated	69	33	47.83%	36.47–59.41	
Clinical status					0.0111
symptomatic	16	11	68.75%	44.40–85.84	
asymptomatic	160	58	36.25%	29.20–43.94	
County of origin					0.0001
Alba	2	1	50%	9.45–90.55	
Bistrița-Năsăud	12	7	58.33%	31.95–80.67	
Cluj	97	49	50.52%	40.74–60.25	
Satu-Mare	45	12	26.67%	15.96–41.04	
Sălaj	20	0	0%	0–16.11	
TOTAL	176	69	39.2%	32.30–46.57	

* Chi-square test.

**Table 2 microorganisms-12-01440-t002:** *E. cuniculi* prevalence based on ELISA testing depending on the family farm of origin.

Family Farm	No. Rabbits Tested	No. Positive Rabbits	Prevalence (%)	CI 95%	*p*
A	3	0	0%	0–56.15	0.0008
B	25	6	24%	11.5–43.43	
C	12	7	58.33%	31.95–80.67	
D	8	7	87.50%	52.91–97.76	
E	12	9	75%	46.77–91.11	
F	5	4	80%	37.55–96.38	
G	28	11	39.29%	23.57–57.59	
H	20	0	0%	0–16.11	
I	19	6	31.58%	15.36–53.99	
J	10	1	10%	1.79–40.42	
K	6	2	33.33%	9.68–70	
TOTAL	148	53	35.81%	28.54–43.80	

## Data Availability

The data presented in this study are available on request from the corresponding author due to owner privacy reasons.

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
