# Peer review of "The Prevalence of Encephalitozoon cuniculi in Domestic Rabbits (Oryctolagus cuniculus) in the North-Western Region of Romania Using Serological Diagnosis: A Preliminary Study"

_microorganisms, 2024, doi:10.3390/microorganisms12071440_

Round 1

Reviewer 1 Report

Comments and Suggestions for Authors

The study by DoboÈ™i reported the prevalence of Encephalitozoon cuniculi in domestic rabbits in the studied area if Romania via  serological diagnosis. Although the data volume is questionable as a research article, the information acquired and discussed is quite important for local epidemic surveillance of E. cuniculi. Following suggestions are for authors' reference to improve their paper.

1) It might be better to verify some of the positive and negative samples using qPCR to support the results of ELISA. It is possible that the false negative/positive results are mixed in the samples.

2) Is it possible to study more in case of the positive samples, e.g., it is located or enriched in some specific areas? And/or the  prevalence in closed related animals, such as dogs, cats and other domestic animals.

3) It is better to summarize/plot the data to the area map so that the readers can understand more about what has been done by authors.

4) Discussion section should be expand a bit more to specify the significance of this study.

5) Limitation of this study must be clearly stated in the last paragraph.

Comments on the Quality of English Language

Minor

Author Response

Dear Reviewer 1,

We appreciate all of the constructive criticisms, useful comments and thoughtful suggestions. All substantive points and suggestions arising from the reviewer have been carefully considered during revision of the original manuscript and implemented (please see below).

All revisions made to the original paper have been highlighted in red color.

A detailed response to all comments from the reviewer is outlined in the accompanying letter indicated by sentences starting with AU:. The original paper has been extensively revised in accordance with the comments and recommendations arising from the peer-review process, and as a result is much improved.

We hope that the revisions would allow the manuscript to be considered acceptable for publication.

Reviewer 1# The study by Doboși reported the prevalence of Encephalitozoon cuniculi in domestic rabbits in the studied area if Romania via serological diagnosis. Although the data volume is questionable as a research article, the information acquired and discussed is quite important for local epidemic surveillance of E. cuniculi. Following suggestions are for authors' reference to improve their paper.

1) It might be better to verify some of the positive and negative samples using qPCR to support the results of ELISA. It is possible that the false negative/positive results are mixed in the samples.

AU: Thank you for your observation. We consider two types of diagnosis methods: serological techniques, such as ELISA, which detect antibodies; molecular genetic techniques, such as different PCR methods, which detect the specific DNA.The diagnosis using molecular genetic techniques will be the object of a future study, followed by comparative studies between all methods.

2) Is it possible to study more in case of the positive samples, e.g., it is located or enriched in some specific areas? And/or the  prevalence in closed related animals, such as dogs, cats and other domestic animals.

AU: This study focuses solely on Encephalitozoon cuniculi detection in rabbits, as they are the main host, therefore prevalence in other species has not been analyzed in this context yet. However, it is a fair point to be taken into consideration for future studies. In regard to the first observation, Table 1. and Table 2. show the seroprevalences based on factors such as age, sex, clinical status, season of sampling and counties of origin, and the seroprevalences in the different family farms analyzed, respectively, which are already presented in our article – pages 5 to 7, lines 203-279.

3) It is better to summarize/plot the data to the area map so that the readers can understand more about what has been done by authors.

AU: Thank you for pointing this out. Therefore, we have revised this by adding a map with the counties that were tested highlighted, together with the number of samples tested in each of them – page 3, Figure 1.

4) Discussion section should be expand a bit more to specify the significance of this study.

AU: We agree with your observation, therefore we expanded the last paragraph of the Discussion section, lines 299-306.

5) Limitation of this study must be clearly stated in the last paragraph.

AU: We agree with this comment, therefore we added a clearer statement in the last paragraph – page 8, lines 315-317.

Reviewer 2 Report

Comments and Suggestions for Authors

The manuscript under title (The prevalence of Encephalitozoon cuniculi in domestic rabbits (Oryctolagus cuniculus) in the North-Western region of Roma nia using serological diagnosis) investigated seroprevalence of E. cuniculi in rabbits. The results of this study are of limited value where the whole study depends upon ELISA kits ready to use for diagnosis of this protozoon in rabbits. There are some suggestions to improve the paper presentation

1.      The main issue is why did the authors used ELISA kits only? Why no molecular study?

2.      The introduction should be include a paragraph about the diagnostic tools for this disease in rabbits specially ELISA

3.      In M&Ms : sampling could be represented by a table

4.      How did the authors detect BCS?

5.      Vaccination for what? May be for some infectious diseases for rabbits or specific vaccine to E. cuniculi?

6.      Serological analysis: what about the replicates of this analysis?

7.      Table 2: from my view of no significant due to the samples are variable and in most of them small like, 3, 5, 6, 8……

8.      Discussion:  lines 141 to 156 introductory may be of benift in the introduction section. The authors repeated the numbers and percentages of the results why? The authors need to support their use ELISA only in this study. The symptomatic rabbits are (11/16), while a symptomatic animals are (58/160), it is difficult to compare these results.

9.      Conclusion is ok. 

Author Response

Dear Reviewer 2,

We appreciate all of the constructive criticisms, useful comments and thoughtful suggestions. All substantive points and suggestions arising from the reviewer have been carefully considered during revision of the original manuscript and implemented (please see below).

All revisions made to the original paper have been highlighted in red color.

A detailed response to all comments from the reviewer is outlined in the accompanying letter indicated by sentences starting with AU:. The original paper has been extensively revised in accordance with the comments and recommendations arising from the peer-review process, and as a result is much improved.

We hope that the revisions would allow the manuscript to be considered acceptable for publication.

Reviewer 2# The manuscript under title (The prevalence of Encephalitozoon cuniculi in domestic rabbits (Oryctolagus cuniculus) in the North-Western region of Romania using serological diagnosis) investigated seroprevalence of E. cuniculi in rabbits. The results of this study are of limited value where the whole study depends upon ELISA kits ready to use for diagnosis of this protozoon in rabbits. There are some suggestions to improve the paper presentation.

  1. The main issue is why did the authors used ELISA kits only? Why no molecular study?

AU: Thank you for your questions. This study focuses solely on the serological diagnosis of Encephalitozoon cuniculi in rabbits, as it is one of the best diagnostic methods in live rabbits for this microorganism, using only blood. The present paper is the first to provide preliminary information of E. cuniculiseroprevalence on the territory of Romania. The reasoning for using ELISA kits was revised in the Discussion section, lines 194-195.

  1. The introduction should be include a paragraph about the diagnostic tools for this disease in rabbits specially ELISA

AU: We agree on this observation, therefore lines 44-68 were revised in the Introduction paragraph, by adding new information as well as the paragraph you suggested we move from the Discussion section here (comment no.8). A new citation [9] was added for this.

  1. In M&Ms : sampling could be represented by a table.

AU: Thank you for your input. In order to avoid repetition of the same data, in the M&M’s section we introduced a reference to Table 1 from the Results section, where all the requested data is provided – page 3, lines 127-129.

  1. How did the authors detect BCS?

AU: BCS (body condition score) was determined by the standards of PFMA. (2013b) Pet Size-O-Meter, which we now also cited in the paper as [11] and lines 112-115 have also been revised.

  1. Vaccination for what? May be for some infectious diseases for rabbits or specific vaccine to E. cuniculi?

AU: In lines 115-118, under the Materials and Methods section, it is stated that the vaccination for which the status was monitored was against myxomatosis and RHD (rabbit hemorrhagic disease, strains 1 and 2). There is currently no vaccine against E. cuniculi, to our knowledge.

  1. Serological analysis: what about the replicates of this analysis?

AU: Thank you for this observation. We clarified the serological analysis process in response to your question, in lines 144-146, explaining that all samples were tested in replicates on the same plate.

  1. Table 2: from my view of no significant due to the samples are variable and in most of them small like, 3, 5, 6, 8……

AU: Thank you for your input. Even though the batches are indeed of a small size, we considered leaving this analysis in the paper, as there was a statistically significant difference in the seroprevalence based on the family farm of rabbit origin. This is a preliminary study, as we added in the title also, which leaves us the path for future studies on larger populations. If you further consider we should eliminate it from our study, please let us know.

  1. Discussion:lines 141 to 156 introductory may be of benift in the introduction section. The authors repeated the numbers and percentages of the results why? The authors need to support their use ELISA only in this study. The symptomatic rabbits are (11/16), while a symptomatic animals are (58/160), it is difficult to compare these results.

AU: We agree with this comment, therefore we have moved that paragraph from the Discussion section to the Introduction section, lines 52-68. The percentages were highlighted again because each of them was individually discussed and compared to other studies.  The use of ELISA was supported by adding lines 194-195. The unequally distributed number of symptomatic vs asymptomatic rabbits is one of the limitations of this study, however we still find it to be relevant information and it proved to have a statistically significant difference in relation to the seroprevalence. This is a preliminary study, as we added now in the title, which indicates that futures studies on larger populations are in plan.

  1. Conclusion is ok.

AU: Thank you for your input. As per another reviewer’s suggestion we added a highlight regarding the limitations of the study in the Conclusion section.

Reviewer 3 Report

Comments and Suggestions for Authors

The article is devoted to the study of the spread of intestinal Encephalitozoon cuniculi in rabbits raised at home. The aim of the work was to study the spread of intestinal Encephalitozoon cuniculi in Romania. The authors claim that this study can help in developing preventive measures for rabbits (reducing the prevalence). The article has both pros and cons. The pros include: the article is sufficiently structured and clearly written; reproducibility of the methodology and its description understandable to readers; a short but informative introduction; informative discussion. The cons include:

1.  In addition to the parameters considered by the authors for rabbits, it is also necessary to take into account the nutrition of the selected animals, as well as the conditions of detention, since this can affect the manifestation of disease symptoms;

2. The list of references is presented mostly by articles published more than 5 years ago;

3. From the introduction it is difficult to conclude whether similar studies have been conducted previously (it is doubtful that the prevalence of intestinal Encephalitozoon cuniculi among rabbits in this area has never been studied, if so, it would be interesting to read the authors' opinion on why this has not been done earlier) without this information it is difficult to conclude on the need to conduct such studies;

4. The article lacks graphic materials (figures); for data expressed as a percentage relative to the general population, it is more informative to construct diagrams.

The article may be re-reviewed after all corrections have been made.

Comments on the Quality of English Language

Minor editing of English language required

Author Response

Dear Reviewer 3,

We appreciate all of the constructive criticisms, useful comments and thoughtful suggestions. All substantive points and suggestions arising from the reviewer have been carefully considered during revision of the original manuscript and implemented (please see below).

All revisions made to the original paper have been highlighted in red color.

A detailed response to all comments from the reviewer is outlined in the accompanying letter indicated by sentences starting with AU:. The original paper has been extensively revised in accordance with the comments and recommendations arising from the peer-review process, and as a result is much improved.

We hope that the revisions would allow the manuscript to be considered acceptable for publication.

Reviewer 3# The article is devoted to the study of the spread of intestinal Encephalitozoon cuniculi in rabbits raised at home. The aim of the work was to study the spread of intestinal Encephalitozoon cuniculi in Romania. The authors claim that this study can help in developing preventive measures for rabbits (reducing the prevalence). The article has both pros and cons. The pros include: the article is sufficiently structured and clearly written; reproducibility of the methodology and its description understandable to readers; a short but informative introduction; informative discussion. The cons include:

  1. In addition to the parameters considered by the authors for rabbits, it is also necessary to take into account the nutrition of the selected animals, as well as the conditions of detention, since this can affect the manifestation of disease symptoms;

AU: Thank you for this observation. As it is true that nutrition and environment conditions can have an impact on initiating the infection with E. cuniculi, these parameters were not recorded for this current study, therefore it would be unfair to categorize the rabbits based on these just by memory. Regarding housing, we can say that pet rabbits were kept inside, either free-range or in enclosure, while the farm rabbits were in a reared household system, kept outside. Please let us know if you would like us to introduce this in our paper.

  1. The list of references is presented mostly by articles published more than 5 years ago;

AU: Two additional references have been added: [4, 9], with new information, in the Introduction section. Besides these, several references from the last 5 years have been cited: [1, 6, 11, 12, 19, 20, 21, 25, 30, 34]. The absence of other more recent studies is explained by the lack of new studies on this topic, more exactly on E. cuniculi in rabbits, and no new or different information compared to older studies we cited. Newer studies focusing on this microorganism in other species are in the literature, that we did not see fitting to citing in this paper.

  1. From the introduction it is difficult to conclude whether similar studies have been conducted previously (it is doubtful that the prevalence of intestinal Encephalitozoon cuniculi among rabbits in this area has never been studied, if so, it would be interesting to read the authors' opinion on why this has not been done earlier) without this information it is difficult to conclude on the need to conduct such studies;

AU: Thank you for your observation. We revised this by adding a new paragraph in the Introduction section, lines 80-85. Hopefully the importance of this study was highlighted.

  1. The article lacks graphic materials (figures); for data expressed as a percentage relative to the general population, it is more informative to construct diagrams.

AU: In response to this comment, we have created a few diagrams that reflect some of the information that is comprised in the main table. However, in our opinion the tables we already have in our paper are clear enough to present the results of our study and diagrams are not exactly suited for the current state, as we only have the “positive/negative” result and no other values in-between that would compare the information better through diagrams. Therefore, if you find the diagrams we created necessary for our paper publication, we suggest adding them as supplementary files (Fig. S1).

Round 2

Reviewer 2 Report

Comments and Suggestions for Authors

The authors replied to my comments

Author Response

Thank you!

Reviewer 3 Report

Comments and Suggestions for Authors

I thank the authors for their attention to the comments and the edits they made.

However, I have one more remark. If the authors indicate that the type of food is not so important for their study, then it is necessary to provide a justification for this statement in the text of the article. In addition, it is necessary to add information about the conditions in which the rabbits are kept.

Author Response

Dear Reviewer,

We appreciate all of the constructive criticisms, useful comments and thoughtful suggestions. All substantive points and suggestions arising from the reviewer have been carefully considered during revision of the original manuscript and implemented (please see below).

All revisions made to the original paper have been highlighted in red color.

A detailed response to all comments from the reviewer is outlined in the accompanying letter indicated by sentences starting with AU:. The original paper has been extensively revised in accordance with the comments and recommendations arising from the peer-review process, and as a result is much improved.

We hope that the revisions would allow the manuscript to be considered acceptable for publication.

With our best regards,

Dr. Anamaria Ioana Paștiu

Reviewer 3# I thank the authors for their attention to the comments and the edits they made. However, I have one more remark. If the authors indicate that the type of food is not so important for their study, then it is necessary to provide a justification for this statement in the text of the article. In addition, it is necessary to add information about the conditions in which the rabbits are kept.

AU: We have added the following paragraph to the Materials section: “Pet rabbits were kept inside the household, either free-range or in an enclosure, with low contact to other rabbits or other animals. Family farm rabbits were mainly kept outside, either in multi-rabbit enclosures or single cages, thus with a higher risk of contamination from other rabbits or species from the surrounding area. This data was not recorded for each individual and taken into analysis due to the variability of this factor, e.g. pet rabbits still having a potential contact with other animals and outside conditions by going into the veterinary facility for medical care.”

AU: We have added the following paragraph to the Discussion section: “The high prevalences cited by other authors [5, 14, 18, 19, 20, 25, 27, 29] highlight the importance that should be put on reducing the contamination risks and increasing the prophylactic measures. Although the type of food consumed by rabbits has been reported to possess a risk for E. cuniculi infection due to potential contamination [4], there are studies that showed that food is not among the risk factors for E. cuniculi infection [31]. Thorough disinfection of the environment and periodic serological testing of rabbits remain the main techniques for diminishing the spread of E. cuniculi in rabbit populations worldwide.”

Thank you for all your support!
